# Sustained SREBP-1-dependent lipogenesis as a key mediator of resistance to BRAF-targeted therapy

Ali Talebi[1], Jonas Dehairs[1], Florian Rambow[2,3], Aljosja Rogiers[2,3], David Nittner[4,5], Rita Derua[6], Frank Vanderhoydonc[1], Joao A.G. Duarte[7,8], Francesca Bosisio[9,10], Kathleen Van den Eynde[9,10], Kris Nys[11], Mónica Vara Pérez[11], Patrizia Agostinis[11], Etienne Waelkens[6], Joost Van den Oord[9,10], Sarah-Maria Fendt[7,8], Jean-Christophe Marine[2,3] & Johannes V. Swinnen [1]

Whereas significant anti-tumor responses are observed in most BRAF$^{V600E}$-mutant melanoma patients exposed to MAPK-targeting agents, resistance almost invariably develops. Here, we show that in therapy-responsive cells BRAF inhibition induces downregulation of the processing of Sterol Regulator Element Binding (SREBP-1) and thereby lipogenesis. Irrespective of the escape mechanism, therapy-resistant cells invariably restore this process to promote lipid saturation and protect melanoma from ROS-induced damage and lipid peroxidation. Importantly, pharmacological SREBP-1 inhibition sensitizes BRAF$^{V600E}$-mutant therapy-resistant melanoma to BRAF$^{V600E}$ inhibitors both in vitro and in a pre-clinical PDX in vivo model. Together, these data indicate that targeting SREBP-1-induced lipogenesis may offer a new avenue to overcome acquisition of resistance to BRAF-targeted therapy. This work also provides evidence that targeting vulnerabilities downstream of oncogenic signaling offers new possibilities in overcoming resistance to targeted therapies.

[1] Laboratory of Lipid Metabolism and Cancer, Department of Oncology, LKI–Leuven Cancer Institute, KU Leuven, 3000 Leuven, Belgium. [2] Laboratory for Molecular Cancer Biology, VIB Center for Cancer Biology, 3000 Leuven, Belgium. [3] Laboratory for Molecular Cancer Biology, Department of Oncology, KU Leuven, 3000 Leuven, Belgium. [4] Histopathology Expertise Center, VIB-KU Leuven Center for Cancer Biology, 3000 Leuven, Belgium. [5] Department of Oncology, KU Leuven, 3000 Leuven, Belgium. [6] Laboratory of Protein Phosphorylation and Proteomics,Department of Cellular and Molecular Medicine, KU Leuven, 3000 Leuven, Belgium. [7] Laboratory of Cellular Metabolism and Metabolic Regulation, Department of Oncology, LKI–Leuven Cancer Institute, KU Leuven, 3000 Leuven, Belgium. [8] Laboratory of Cellular Metabolism and Metabolic Regulation, VIB Center for Cancer Biology, 3000 Leuven, Belgium. [9] Translational Cell and Tissue Research, Department of Imaging and Pathology, KU Leuven, Belgium. [10] Department of Pathology, UZ Leuven, 3000 Leuven, Belgium. [11] Laboratory of Cell Death Research & Therapy, Department of cellular and molecular medicine, KU Leuven, 3000 Leuven, Belgium. These authors contributed equally: Ali Talebi, Jonas Dehairs. These authors jointly supervised this work: Jean-Christophe Marine, Johannes V. Swinnen. Correspondence and requests for materials should be addressed to J.V.S. (email: j.swinnen@kuleuven.be)

While targeted approaches are revolutionizing the treatment of cancer, the management of both intrinsic and acquired therapy resistance remains a major limitation. This is exemplified by the unprecedented, but transient, anti-tumor responses seen in patients with BRAF[V600E]-mutant malignant melanoma exposed to agents that selectively inhibit oncogenic BRAF[1,2]. Many of these patients show almost complete remission in response to such targeted agents, however, therapy resistance eventually develops in ~80% of all cases[3–5].

Many genomic and non-genomic mechanisms have been described, all leading to re-activation of the MAPK- and/or PI3K-signaling pathways[6–8]. Moreover, different mutational events can be selected in distinct drug-resistant clones from the same patient[9] and even co-occur within the same lesion[10]. These findings have highlighted the need to improve effectiveness of treatment, by for instance, the co-targeting of other essential cancer vulnerabilities and/or key mediators of MAPK signaling itself.

One of the pathways that is emerging as a central player in multiple oncogenic processes and that functions downstream of a multitude of oncogenic signal transduction pathways is de novo lipogenesis. Accordingly, this pathway is specifically activated in many cancers[11–14], in part through induction of the transcription factor Sterol Regulatory Element Binding Protein (SREBP-1), a master regulator of lipogenesis[15–20]. Aberrant activation of the lipogenic pathway in cancer is required for the synthesis of phospholipids, which function as essential building blocks of membranes and that support cell growth and proliferation[21,22]. As this pathway mainly produces saturated and mono-unsaturated fatty acids, an increase in the proportion of these lipids in the cellular membrane composition of cancer cells is often observed[23–26]. Importantly, saturated and mono-unsaturated fatty acids are less prone to lipid peroxidation, thereby providing a survival advantage to cancer cells, particularly those exposed to oxidative stress[26].

Here, we show that the lipogenic pathway is a key mediator of oncogenic BRAF and that its constitutive activation, which is mediated by SREBP-1, contributes to therapy resistance. Our findings support the use of SREBP-1 inhibitors in a novel combinatorial approach to overcome resistance to BRAF[V600E]-targeted therapy.

## Results

**De novo lipogenesis is inhibited by BRAF[V600E]-targeted therapy.** As in many cancers, there is evidence that de novo lipogenesis is activated in melanoma[27,28]. We reasoned that ectopic MAPK-activation may be one key triggering event of such activation. To test this possibility, we assessed the impact of BRAF inhibition on lipid metabolism. We exposed BRAF-mutant, therapy-sensitive, melanoma cell lines (M249 and A375) to vemurafenib and profiled their transcriptome by RNA-seq. Ingenuity pathway analysis (IPA) identified fatty acid metabolism as one of the most affected pathways by the treatment (Fig. 1a). Consistently, expression of key lipogenic enzymes such as ATP citrate lyase (ACLY), acetyl-CoA carboxylase-1 (ACACA), and fatty acid synthase (FASN) were consistently decreased (Fig. 1b, Supplementary Fig. 1a). Alterations in the expression of these enzymes by mutant BRAF inhibition was confirmed by RT-qPCR on an extended panel of therapy-sensitive BRAF[V600E] parental and isogenic cell lines that have acquired resistance to vemurafenib through diverse mechanisms (Supplementary Table 1). These include Raf-kinase flexibility in MAPK signaling and in increased IGF-1R/PI3K signaling (451lu R)[29], enhanced RTK signaling (M229 R and M238 R) and secondary acquisition of oncogenic NRAS[Q61K] (M249 R)[30]. Whereas vemurafenib decreased the expression of lipogenic enzymes in all sensitive BRAF-mutant cell lines, this

was not seen in normal neonatal human epidermal melanocytes (NHEM) and in the therapy-resistant lines (Fig. 1c, Supplementary Fig. 1b). If anything, the opposite effect was observed in the vemurafenib-resistant cells. Direct measurement of the overall rate of lipogenesis by assessing [14]C-acetate incorporation into lipids confirmed an overall increase in lipogenesis in melanoma cell lines compared to NHEM (Fig. 1d). A marked decrease in de novo lipogenesis was observed in all BRAF[V600E] therapy-sensitive, but not resistant, cell lines upon vemurafenib exposure. These findings were further corroborated by isotopomer spectral analysis, a method that measures fatty acid biosynthesis rates by measuring the fraction of de novo synthesized palmitate. In general, there was a marked decrease in the fraction of de novo synthesized palmitate in therapy-sensitive lines. In contrast, vemurafenib did not cause any decrease in palmitate synthesis in some therapy-resistant cells or induced only a moderate reduction in others (Supplementary Fig. 2). We conclude that lipogenesis is sustained in therapy-resistant cells when compared to therapy-sensitive cells upon vemurafenib treatment. Notably, lipid uptake, cholesterol synthesis rate or cholesterol uptake were not affected in any of the conditions and cell lines, indicating that vemurafenib predominantly affects de novo fatty acid biosynthesis (Supplementary Fig. 3a–c).

De novo lipogenesis mainly produces saturated and mono-unsaturated fatty acids with phospholipids as major end product[24–26]. Consistently, mass spectrometry-based phospholipidome analysis revealed that inhibition of oncogenic BRAF in the therapy-sensitive lines caused an increase in the proportion of poly-unsaturated membrane phospholipid species at the expense of saturated and mono-unsaturated phospholipids. These are typical changes observed upon lipogenesis inhibition[26] (Fig. 1e, f). Such a shift was either absent or less pronounced in the therapy-resistant lines and in NHEM. Taken together, these findings indicate that inhibition of oncogenic BRAF inhibits de novo lipogenesis and thereby enhances membrane poly-unsaturation.

**BRAF[V600E]-induced lipid metabolism is mediated by SREBP-1.** The selected lipogenic enzymes, the expression of which are downregulated upon oncogenic BRAF inhibition, are well-established transcriptional targets of SREBP-1. We therefore examined whether activity of SREBP-1 itself may be decreased by vemurafenib. Because SREBP-1 is synthesized as an inactive precursor, which is activated upon proteolytic cleavage[31] (Fig. 2a), we used western blot analysis to assess the protein levels of both full-length and mature SREBP-1. Vemurafenib caused a decrease in the levels of the mature form of SREBP-1 in all BRAF[V600E]-therapy-sensitive, but not (or less so) in resistant cell lines (Fig. 2b). In contrast, the non-processed form was either unaffected or increased. To further substantiate this finding, we exploited the paradoxical activation of the MAPK pathway by vemurafenib in two NRAS mutant cell lines (M202 and M207)[32–34]. Vemurafenib resulted in both an expected increase in the levels of pMEK 1/2 proteins and of mature SREBP-1 (Fig. 2c). In addition, over-expression of a BRAF[V600E]-encoding plasmid in these BRAF wild-type cells further support the ability of oncogenic BRAF to induce SREBP-1 processing (Fig. 2d). RT-qPCR analysis of the transcripts encoding SREBP-1a and SREBP-1c showed a decreased expression upon vemurafenib exposure in 451lu and A375, but not in the other cell lines (Supplementary Fig. 4). As SREBP expression is subject to autoregulation, these findings indicate that vemurafenib acts, at least in part, at the level of SREBP maturation. Note, however, that more direct transcriptional effects on the regulation of SREBF-1 transcription cannot be fully excluded. To further substantiate these findings, we looked at the effect of vemurafenib on the cellular distribution of

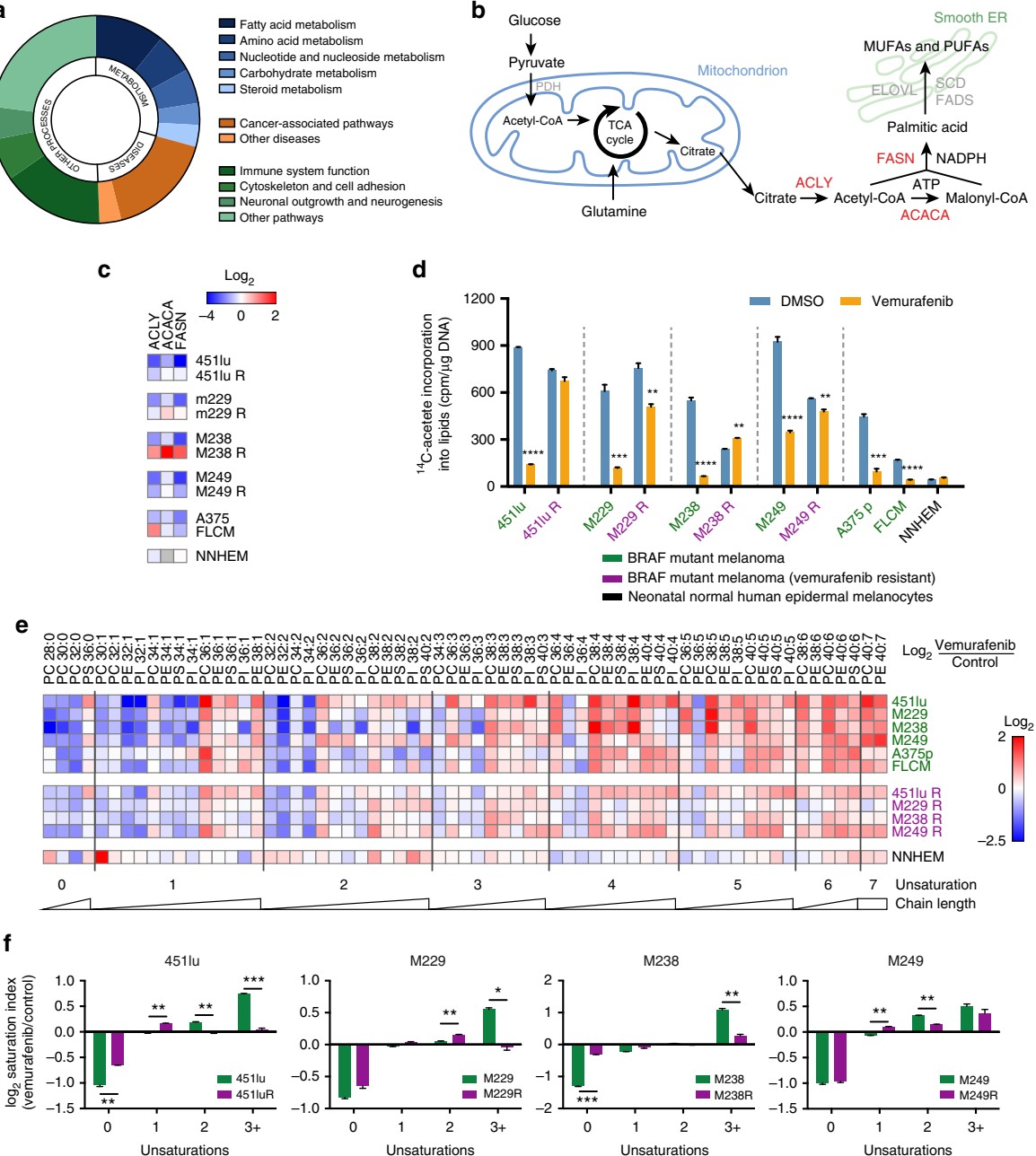

**Fig. 1** De novo lipogenesis is inhibited by vemurafenib in therapy-sensitive cells, but remains activated in therapy-resistant ones. **a** Significantly altered (fold change of at least 1.5 and $p < 0.05$) pathways as indicated by ingenuity pathway analysis (IPA) of RNA-seq of vemurafenib (5 µM) treated M249 and A375 cells ($n = 3$). **b** Vemurafenib affects major components of the fatty acid synthesis pathway (fatty acid synthase, ATP citrate lyase and acetyl-CoA carboxylase). **c** The effects of vemurafenib treatment on the mRNA levels of FASN, ACLY, and ACACA in therapy-sensitive versus resistant BRAF-mutant cell lines and in NHEM ($n = 3$). **d** Vemurafenib or vehicle treated cells were assayed for their ability to incorporate $^{14}$C-acetate into lipids. Data are represented as mean ± s.e.m. The significance was determined with an unpaired $t$-test and compares vemurafenib-treated cells to their matching controls ($n = 3$). (*$p < 0.05$, **$p < 0.01$, ***$p < 0.001$, ****$p < 0.0001$). **e** Heatmap of $\log_2$ ratios of the abundance of phospholipid species in vemurafenib-treated cells over vehicle treated cells. Species are indicated by their total number of fatty acid carbons, followed by a colon and the total number of unsaturations ($n = 3$). **f** $\log_2$ ratios of the vemurafenib-induced changes in saturation index. The saturation index was calculated by summing the species with the same level of unsaturations

SREBP-1. To this end, we generated a transcriptionally inactive recombinant full-length SREBP-1 construct with an N-terminal HA-tag and a C-terminal myc-tag, allowing visualization of the active and inactive forms of the protein (Fig. 2e). Western blotting established that the transgene is processed in a manner that is indistinguishable from endogenous SREBP-1 (Supplementary Fig. 5a). Similarly to endogenous mSREBP-1, which localizes to the nucleus, the processed HA-tagged exogenous protein was detected in the nuclei of untreated melanoma cells. In contrast, both HA- and myc-tagged proteins co-localized in the ER–Golgi, indicating that proteolytic activation is halted in vemurafenib-treated cells (Fig. 2f, Supplementary Fig. 5b). Taken together, oncogenic BRAF targeting inhibits the processing and activation of SREBP-1 in therapy-sensitive, but not therapy-resistant, melanoma cells and this effect is, by and large, mediated by a post-translational mechanism.

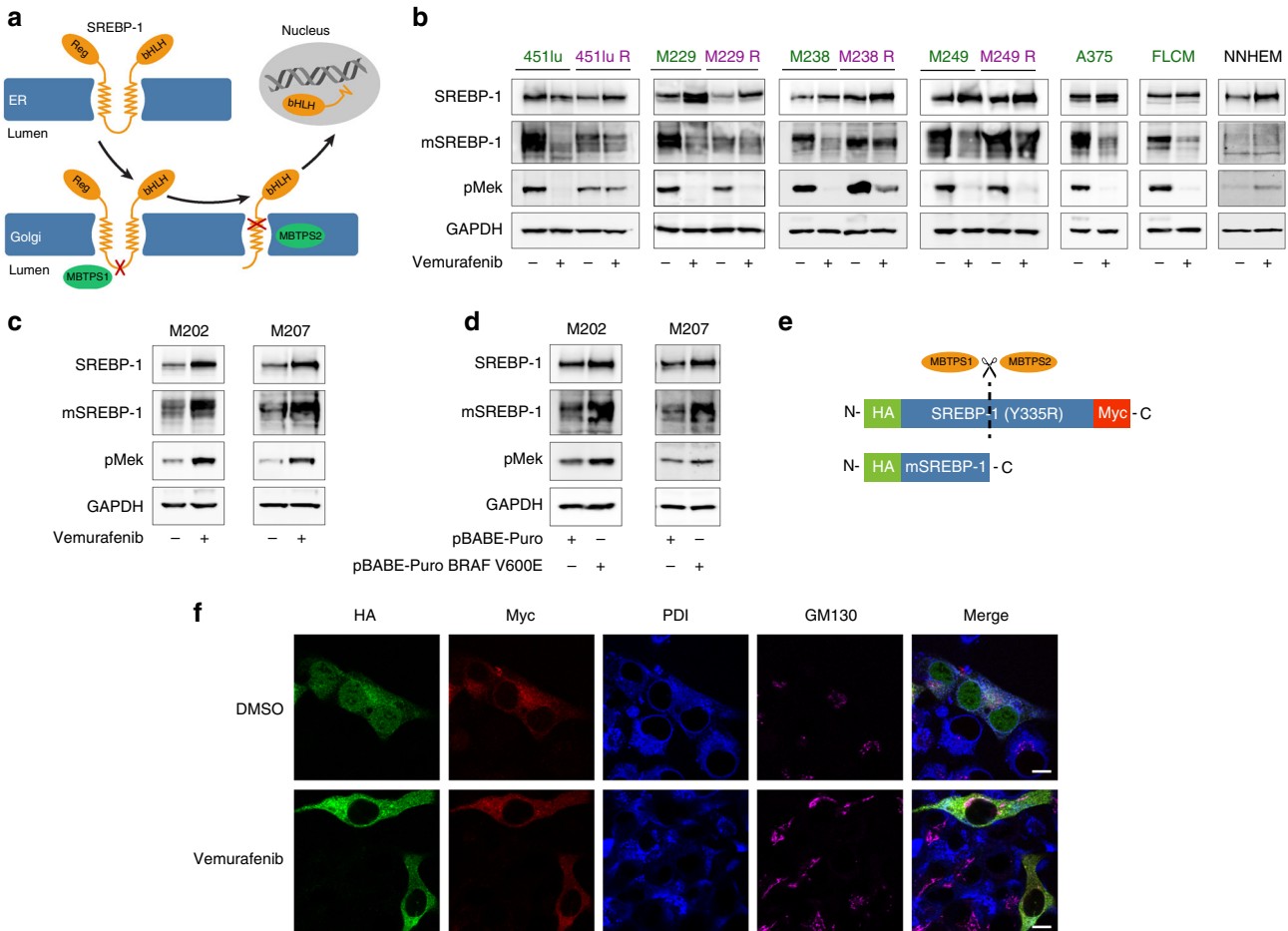

**Fig. 2** Effects of vemurafenib on lipid metabolism involves SREBP-1. **a** Diagram illustrating the processing of SREBP: through SCAP interaction, SREBP is transported from the ER to the Golgi where it is consecutively cleaved by two proteases, yielding transcriptionally active, mature SREBP (mSREBP). **b**–**d** Western analysis of SREBP-1 (full-length and mature) and pMEK (Ser217/221). GAPDH is used as a loading control. **b** Western blot analysis for mSREBP levels in response to vemurafenib treatment in sensitive and resistant cells. **c** Western blot analysis of BRAF wild-type (but NRAS mutant) melanoma cells M202 and M207 treated with vemurafenib (5 μM). **d** Western blot analysis of BRAF wild-type, NRAS mutant melanoma cells, 72 h after transfection with pBABE-Puro BRAF$^{V600E}$. **e** Diagram illustrating the HA-SREBP-1(Y335R)-myc construct. The Y335R mutation renders the construct transcriptionally inactive. **f** Confocal analysis of 451lu cells expressing HA-SREBP-1(Y335R)-myc. A PDI antibody was used to mark the ER and GM130 was used for *cis*-Golgi. Scale bar indicates 25 μM

Interestingly, BRAF inhibition only induced a moderate decrease in mSREBP-1 levels and did not significantly affect lipogenesis in therapy-resistant cells (Fig. 2b). Since alternative activation of the ERK pathway is a common contributor to therapy resistance and a known regulator of SREBP[20,35,36], we treated the therapy-resistant cell line 451lu R with the MEK inhibitor trametinib (Supplementary Fig. 6). As expected, these cells maintained high levels of pMEK upon vemurafenib treatment (Fig. 2b). Interestingly, MEK inhibition substantially decreased the levels of mSREBP-1 in these cells (Supplementary Fig. 7). Consistently, expression of well-established mSREBP-1 downstream targets, such as ACLY, ACACA, and FASN, was also reduced (Supplementary Fig. 8). These findings indicate that re-activation of the ERK pathway contributes to sustained SREBP-1 activity in therapy-resistant melanoma cells.

**Sustained SREBP-1 activity maintains lipogenesis in therapy-resistant cells**. To further assess the role of SREBP-1 in the changes in lipid metabolism evoked by vemurafenib, we inhibited SREBP-1 in 451lu R cells using pharmacological and genetic approaches. We used two small-molecule inhibitors, betulin and

fatostatin, which inhibit the trafficking of SREBP to the Golgi, and thereby its proteolytic activation through distinct mechanisms[37,38]. Exposure of 451lu and 451lu R cells to these inhibitors induced the expected dose-dependent decrease in the levels of mature SREBP-1; an effect that was more pronounced in the therapy-sensitive cell line (Supplementary Fig. 9a). Phospholipidomic analysis revealed that chemical inhibition of SREBP-1 dose-dependently depleted mono-unsaturated and fully saturated phospholipid species and increased membrane poly-unsaturation, partially recapitulating the effect of BRAF inhibition on the therapy-sensitive cell line (Fig. 3a). Furthermore, these effects were further enhanced with the addition of vemurafenib, whereby the phospholipidome of the resistant line under SREBP-1 inhibition and vemurafenib closely resembled that of the sensitive line in response to vemurafenib (Fig. 3a).

To corroborate these data using a genetic approach, we generated a heterozygous and two homozygous SREBF-1 knock-out clones from the therapy-resistant cell line 451lu R using CRISPR-Cas9. Both Sanger sequencing and western blot analysis confirmed partial or full SREBF-1 deletion in the heterozygous and homozygous mutant cells, respectively (Supplementary Fig. 9b, c)[39]. Similar changes in the lipid membrane composition

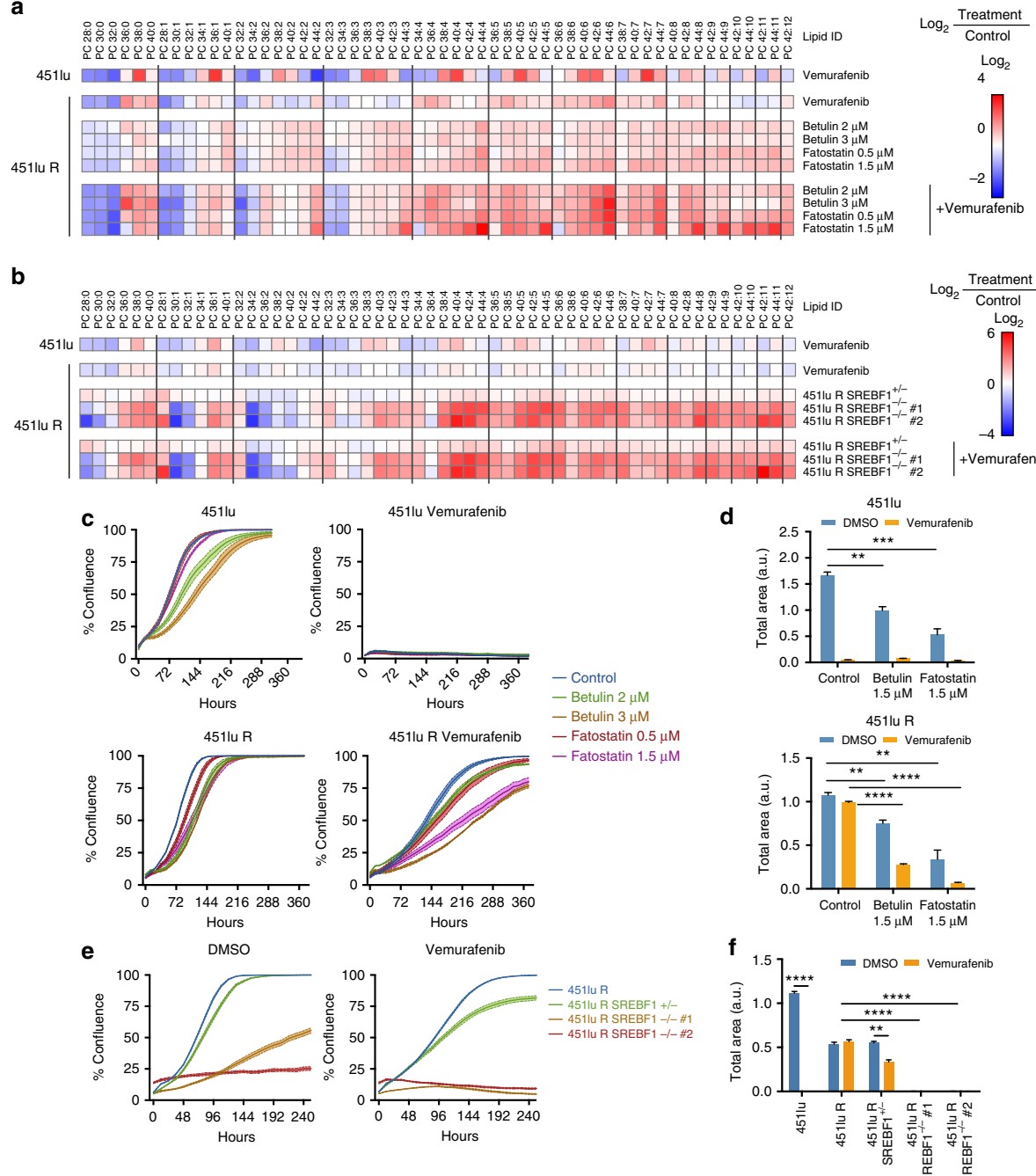

**Fig. 3** SREBP-1 contributes to membrane saturation and therapy resistance. **a** Lipidomic (phosphatidylcholine) profiles of 451lu and 451lu R cells treated with vemurafenib (5 μM) and SREBP inhibitors fatostatin and betulin (*n* = 3). **b** Lipidomics of hetero- and homozygous CRISPR-Cas9 knockouts of SREBF-1 in combination with vemurafenib treatment (*n* = 3). **c** Proliferation curves of 451lu and 451lu R cells treated with vemurafenib and fatostatin or betulin (phase contrast density measured by the Incucyte system, raw images and bar graph are shown in Supplementary Figs. 10, 11) (*n* = 4). **d** 451lu and 451lu R cells were treated with SREBP inhibitors and vemurafenib and were assayed for their ability to form colonies in soft agar, raw images and bar graph are shown in Supplementary Fig. 12 (*n* = 3). **e** Proliferation curves of 451lu R SREBP KO cells, raw images and bar graph are shown in Supplementary Figs. 13, 14 (*n* = 4). **f** SREBF-1 KO 451lu R cells were assayed for their ability to form colonies in soft agar in presence or absence of vemurafenib, raw images and bar graph are shown in Supplementary Fig. 15 (*n* = 3). All data are represented as mean ± s.e.m. (*\*p* < 0.05, \*\**p* < 0.01, \*\*\**p* < 0.001, \*\*\*\**p* < 0.0001)

to the ones observed upon pharmacological inhibition were observed in the heterozygous SREBF-1 KO clone, and to an even greater extent, in homozygous KO clones (Fig. 3b). Since inhibition of SREBP-1 activation resulted in membrane lipid changes that closely mimicked the effects of vemurafenib, we concluded that SREBP-1 is the major mediator of BRAF inhibition-dependent lipid metabolism rewiring.

**Inhibition of SREBP-1 re-sensitizes resistant cells to BRAF targeting therapy.** In order to investigate whether vemurafenib-induced processing of SREBP-1 contributes to its anti-tumor response, we assessed the ability of melanoma cells to grow in both 2D and 3D cultures in response to vemurafenib upon pharmacological or genetic inactivation of SREBP-1. Vemurafenib potently inhibited cell proliferation of the therapy-

sensitive, but not the therapy-resistant line (Fig. 3c, Supplementary Figs. 10, 11). Proliferation of the therapy-resistant cells was strongly inhibited upon combined exposure to vemurafenib and SREBP-1 inhibitors. Similarly, SREBP-1 inhibitors synergized with vemurafenib in inhibiting the ability of therapy-resistant cells to form colonies (Fig. 3d, Supplementary Fig. 12). Genetic ablation of SREBF-1 inhibited the rate of cell proliferation in 2D cultures of therapy-resistant cells when compared to the resistant parental cell line. In contrast, inactivation of only one SREBF-1 allele had no effect on cell proliferation (Fig. 3e, Supplementary Figs. 13, 14). The rate of proliferation of these cells decreased significantly upon exposure to vemurafenib. Similarly, these heterozygous SREBF-1 knockout cells were able to form colonies in 3D cultures; an ability that was reduced in the presence of vemurafenib (Fig. 3f, Supplementary Fig. 15).

In order to assess whether sustained SREBP-1 processing also mediates therapy resistance to vemurafenib in BRAF$^{V600E}$ mutant cells that are endogenously resistant to BRAF targeting therapy, we treated M233[40] with betulin and fatostatin. Both betulin and fatostatin treatment further sensitized the cell line to vemurafenib (Supplementary Fig. 16).

In order to assess whether our results directly reflect the effects of BRAF inhibition and are not off-target effects of vemurafenib, in parallel we treated 451lu and 451lu R cells with dabrafenib. Dabrafenib caused a decrease in both $^{14}$C-acetate incorporation into lipids and mature SREBP-1 protein levels in 451lu cells, but not in 451lu R cells (Supplementary Fig. 17). Furthermore, combined betulin and fatostatin treatment with dabrafenib inhibited proliferation in 451lu R cells, mirroring the effects of vemurafenib (Supplementary Fig. 18).

Since combined BRAF and MEK inhibitor treatment is a standard of care for BRAF-mutant melanoma patients, we further assessed the cell proliferative response to SREBP inhibition in A101D BMR and D10 BMR. These cell lines are resistant to both dabrafenib and trametinib (A101D BMR is partially resistant and retains some level of sensitivity). Here we show that SREBP inhibition by either betulin or fatostatin sensitized the cells to combined BRAF and MEK inhibition (Supplementary Figs. 19–22).

Taken together, these data indicate that SREBP-1 contributes to the anti-tumor response induced by BRAF inhibition and that SREBP-1 inhibition sensitizes therapy-resistant melanoma cells to MAPK-targeting therapy.

**SREBP-1 protects vemurafenib-resistant cells from lipid per-oxidation**. The findings described above predict that SREBP-1-mediated therapy resistance is a consequence of enhanced membrane lipid saturation and, consequently, decreased lipid peroxidation. To test this hypothesis, we mimicked the effect of SREBP-1 inhibition on the lipid composition of therapy-resistant cells by treatment with the poly-unsaturated fatty acids linoleic and linolenic acid (PUFA). When combined with vemurafenib, PUFA addition slightly attenuated the growth of the cultures. Conversely, supplementing cells with a final product of lipogenesis, oleic acid, slightly enhanced proliferation of cells under vemurafenib treatment. Both of these effects were significantly enhanced upon pharmacological or genetic inactivation of SREBP-1 indicating that membrane lipid saturation contributes to SREBP-1-mediated resistance to BRAF inhibition (Fig. 4a, Supplementary Figs. 23, 24).

We have previously reported that membrane saturation protects cancer cells from ROS- and chemotherapy-induced cell death[26]. Here, we observed that both fatostatin and vemurafenib increase the levels of mitochondrial ROS independently, and that a combination of the two leads to an additional effect (Fig. 4b). In addition, ROS levels were either further enhanced or decreased by

addition of PUFAs and oleic acid, respectively. Under combined vemurafenib and fatostatin treatment, addition of oleic acid reduced the levels of mitochondrial ROS to levels observed in cells treated with vemurafenib alone (Fig. 4b). ROS has been linked to membrane lipid peroxidation, which results in the accumulation of toxic by-products[41]. Since poly-unsaturated lipids are more prone to lipid peroxidation than saturated lipids, we next measured lipid peroxidation by measuring levels of cellular malondialdehyde (MDA), which is a direct by-product of lipid hydro-peroxide degradation. Whereas generally low in melanoma cells, levels of MDA increased upon exposure to fatostatin and to an even greater extent upon vemurafenib treatment (Fig. 4c). Combined fatostatin and vemurafenib treatment resulted in a further increase in MDA levels, which is directly in line with the levels of membrane lipid poly-unsaturation incurred by the treatments (Figs. 3a and 4c). The levels of MDA under combined therapy increased further with addition of exogenous PUFA and decreased to levels seen under vemurafenib alone upon addition of exogenous oleic acid.

Interestingly, steady state metabolomics analysis of cellular AMP, ADP, and ATP indicated that cellular ATP levels or energy charge are not markedly altered by combined BRAF and SREBP-1 inhibition (Supplementary Fig. 25). Metabolomics analysis of NAD, NADH, NADP, NADPH, GSH, and GSSG ratios revealed that combined SREBP-1 and BRAF inhibition significantly lowers the cell antioxidant potential.

To corroborate our findings that SREBP-1 inhibition sensitizes cells to vemurafenib through lipid peroxidation, we supplemented cells with the antioxidants alpha-tocopherol, ferrostatin, and N-acetyl-cysteine (NAC). Under combined SREBP-1 and oncogenic BRAF inhibition, addition of antioxidants partially rescued cell proliferation (Fig. 4d, Supplementary Figs. 26, 27). To show that lipid poly-unsaturation and lipid peroxidation also plays a role in vemurafenib response in drug sensitive cells, we treated therapy-sensitive cell lines M229 and 451lu with either betulin or fatostatin and found that these compounds further enhance the cytostatic effects of the BRAF-inhibitor (Supplementary Figs. 28, 29). Furthermore, we show that treatment with alpha-tocopherol, in part, rescued the cytostatic effects of vemurafenib in the M229 cell line, and alpha-tocopherol, NAC and ferrostatin treatment rescued the proliferation of 451lu cells (Fig. 4e and Supplementary Figs. 30, 31). We conclude that SREBP-1 inhibition sensitizes cells to vemurafenib, at least partly, though alterations of membrane poly-unsaturation and, thereby, lipid peroxidation.

**SREBP-1 inhibition sensitizes melanoma to vemurafenib in vivo**. To assess the therapeutic potential of these findings we investigated the impact of SREBP-1 inhibition in an in vivo pre-clinical BRAF$^{V600E}$-mutant melanoma model. We chose PDX MEL006, which has been extensively characterized previously and was shown to poorly respond to BRAF inhibitors alone[42,43]. Mouse cohorts were treated blindly with either vehicle, vemurafenib alone, fatostatin alone or a combination of vemurafenib and fatostatin. Fatostatin treatment alone inhibited tumor growth more potently than vemurafenib. Importantly, combined vemurafenib/fatostatin co-treatment had a greater anti-tumor effect than any of the monotherapy regimens (Fig. 5a–c). Phospholipidomic analysis of the various treated melanoma lesions revealed a correlation between the changes in the poly-unsaturation of phospholipids and anti-tumor growth response (Fig. 5d), whereby membrane poly-unsaturation was synergistically enhanced by the combination treatment. MDA analysis revealed that whereas fatostatin or vemurafenib treatment alone did not significantly increase lipid peroxidation, the combined vemurafenib/fatostatin treatment greatly enhanced

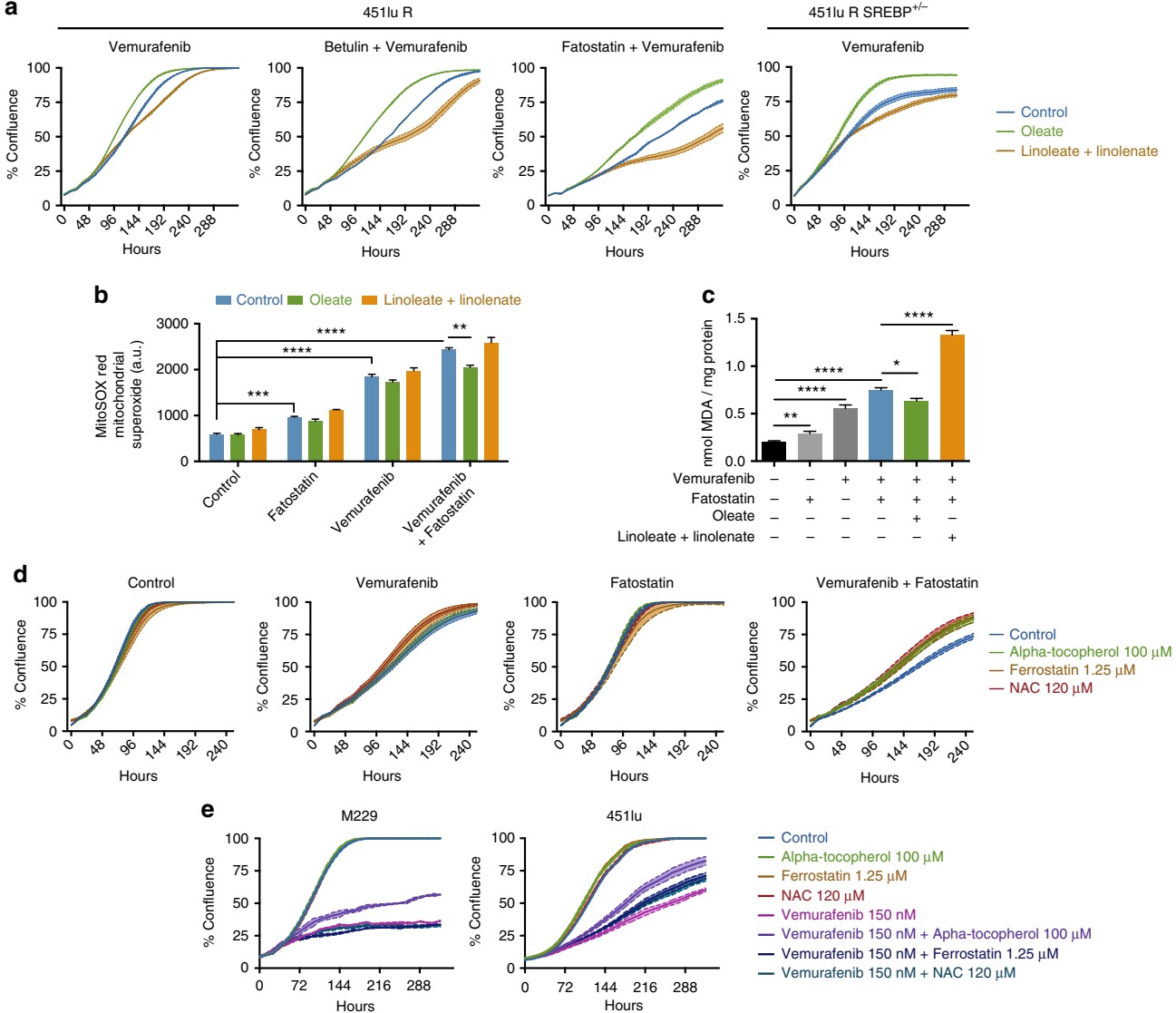

**Fig. 4** Re-sensitization to vemurafenib involves lipid peroxidation. **a** Oleate (20 μM) or a mixture of linoleate (10 μM) + linolenate (10 μM) was added to the culture medium of 451lu R SREBP$^{+/-}$ cells and 451lu R cells treated with either vemurafenib alone or a combination with fatostatin or betulin, raw images and bar graph are shown in Supplementary Figs. 23, 24 ($n = 3$). **b** Mitochondrial superoxide levels in 451lu R cells treated with vemurafenib and fatostatin; medium was supplemented with oleate or linoleate + linolenate ($n = 3$). **c** MDA levels normalized to protein content in 451lu R cells ($n = 5$). **d** Effect of alpha-tocopherol, ferrostatin and NAC supplementation on proliferation curves of 451lu R cells treated with vemurafenib + fatostatin, raw images and bar graph are shown in Supplementary Figs. 26, 27 ($n = 4$). **e** Proliferation curves of the therapy-sensitive cell lines M229 and 451lu treated with vemurafenib, alpha-tocopherol, ferrostatin, and NAC (phase contrast density measured by the Incucyte system) raw images and bar graph are shown in Supplementary Figs. 30, 31 ($n = 3$). All data are represented as mean ± s.e.m. (*$p < 0.05$, **$p < 0.01$, ***$p < 0.001$, ****$p < 0.0001$)

lipid peroxidation (Fig. 5e). In the Mel006 tumor treated with a combination of dabrafenib and trametinib, an increase in MDA was found shortly after the start of treatment and, to a lesser extent, after establishment of resistance (Supplementary Fig. 32).

Taken together, combined fatostatin and vemurafenib therapy enhanced therapy response in vivo and increased membrane lipid poly-unsaturation and lipid peroxidation. These data support the concept of a novel combinatorial approach to overcome therapy resistance in BRAF$^{V600E}$ mutant melanoma models.

## Discussion

Resistance to targeted therapy represents a major clinical challenge. This is partly a consequence of the fact that most therapeutic targets to date, including BRAF$^{V600E}$, act in the proximal part of their signal transduction cascade. This offers multiple opportunities for cancer cells to bypass drug response through for instance the acquisition of mutation(s) that reactivate the pathway downstream (e.g., by MEK mutation). An attractive strategy to overcome therapy resistance is therefore the identification and exploitation of vulnerabilities, which are activated by and act downstream of such oncogenic pathways. Metabolic pathways are of particular interest in this context as they often rely on a few essential enzymes, are frequently rewired in cancer cells, provide essential survival/adaptive capabilities and can easily be pharmacologically targeted. Here, we identified the lipogenic transcription factor SREBP-1 as a key downstream target of oncogenic BRAF signaling. We have shown that sustained lipogenesis through the maintenance of active SREBP-1 is a key feature of therapy resistance to vemurafenib in BRAF-mutant

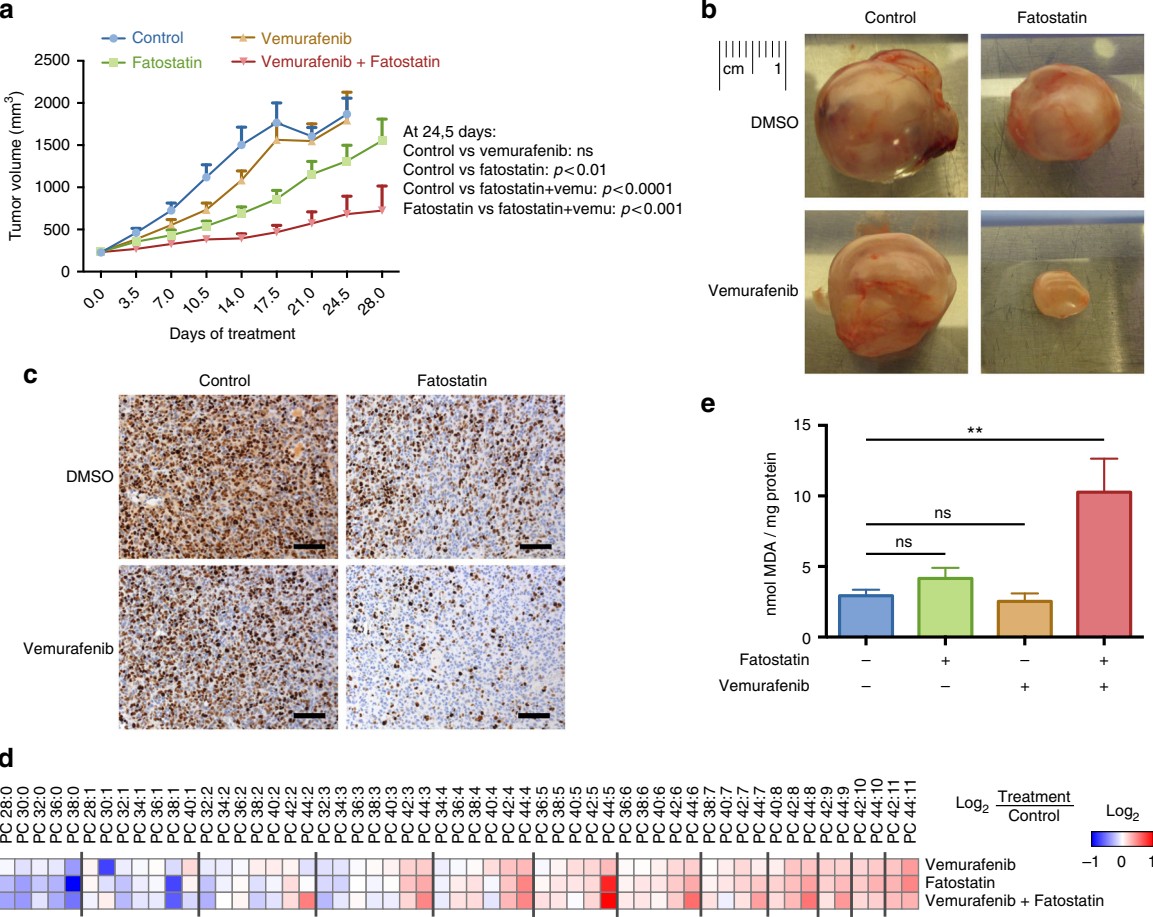

**Fig. 5** SREBP-1 protects vemurafenib-resistant cells from lipid peroxidation. BRAF$^{V600E}$-mutant PDX tumors (mel6 model) were transplanted into nude mice (NMRI-Fox1nu) which were blindly administered vehicle ($n = 8$), fatostatin (20 mg/kg) ($n = 10$), vemurafenib (20 mg/kg) ($n = 10$) or a combination of vemurafenib and fatostatin ($n = 10$) (daily by oral gavage). **a** Tumor size was measured blindly with digital calipers every 3 days. The tumor pictures in (b) show the tumors that represent the median of their respective cohort. **c** Expression of Ki67 protein in PDX tumors that represent the median of their respective cohort. Scale bars indicate 100 μm. **d** Lipidomics of PDX tumor homogenate. **e** MDA quantification of PDX tumor homogenate. The data represent the average over all mice for each cohort compared to the control. Data are represented as mean ± s.e.m. (**$p < 0.01$)

melanoma, and that inhibition of SREBP-1 sensitizes melanoma to targeted therapy.

Critically, the addiction of therapy-resistant melanoma cells to SREBP-1 is independent of the mechanisms exploited by the cancer cells to overcome drug response. We observed similar effects in cells that acquired resistance acquisition of through RTK upregulation (M229 R, M238 R), NRAS mutation (M249 R) or enhanced IGF1/PI3K signaling and RAF kinase flexibility (451lu R). In all sensitive BRAF-mutant models, vemurafenib caused a decrease in lipogenesis and attenuated the processing and thereby the activity of SREBP-1. This was not seen in therapy-resistant models, which all showed high levels of lipogenesis even in the presence of the inhibitor. Together with our observation that pharmacological or genetic inactivation of SREBP-1 in resistant cells attenuates cell proliferation and sensitizes to vemurafenib (irrespective of the escape mechanism), these findings indicate that SREBP-1-mediated lipogenesis is a central pathway acting downstream of mutant BRAF.

Previous work has shown that SREBP-1 is activated through several mechanisms including, regulation by the PTEN/PI3K/Akt/mTOR pathway[20,44], p53[19,45], modulation of MAPK signaling by KRAS[46] and by direct SREBP phosphorylation by Erk1/2[47,48]. Our work identifies mutant BRAF as another key modulator of SREBP-1 processing and function.

Consistent with our findings, SREBP and its downstream targets are highly expressed in many cancers[49]. Importantly there is a growing body of evidence showing that SREBP-1-dependent activation of lipogenesis is required for tumor growth in multiple models, including in prostate cancer[50] and EGFR-dependent glioma[15,18]. SREBP-1 was shown to promote adhesion-independent growth[44] and cell proliferation[51,52,53,54], including growth factor-independent proliferation[55].

Similarly, proteins involved in the post-translational processing of SREBPs have also been linked to oncogenic potential in multiple models. SCAP modification and inhibition inhibit tumor growth through SREPBs[50,56–58]. Consistently, expression levels of SCAP inversely correlate with overall survival in multiple cancers in TCGA cohorts. Taken together, these data strongly support a pro-oncogenic role for SREBP processing in multiple cancers.

These data are in line with the well-established necessity of cancer cells to adapt their metabolism to their increased need of building blocks. Activation of SREBP-1 and enhanced ability to generate lipids in a cell-autonomous manner is thought to be required to sustain rapid tumor cell proliferation[59,60]. Earlier findings from our team have shown that lipogenesis also contributes to resistance to cell death by altering membrane lipid composition and susceptibility to lipid peroxidation[26]. Here we provide evidence that lipogenesis driven by oncogenic BRAF

signaling promotes resistance to targeted therapy. In therapy-sensitive cells, inhibition of oncogenic BRAF decreases membrane lipid saturation. Inhibition of SREBP-1 mimics this effect in therapy-resistant cells, especially upon BRAF therapy. We further show that this effect is a consequence of increased cellular ROS and lipid peroxidation. In therapy-resistant cells, vemurafenib resulted in a substantial increase in mitochondrial ROS and lipid peroxidation. This is consistent with the well-established ability of vemurafenib to induce ROS production[61,62]. Similarly, although to a lesser extent, SREBP inhibition also increased mitochondrial ROS levels. Combined inhibition of SREBP and oncogenic BRAF further enhanced ROS and lipid peroxidation, which can either be rescued or enhanced by exogenous addition of oleate or PUFA, respectively. Furthermore, addition of the antioxidant NAC results in a rescue of cell proliferation under combined therapy.

Importantly, SREBP inhibition enhanced the efficacy of vemurafenib in a pre-clinical PDX model of melanoma, emphasizing the clinical relevance of these findings. Our data support the growing interest in lipogenesis inhibition as a novel anti-neoplastic strategy and ongoing efforts aimed at identifying new classes of SREBP inhibitors, including those that interfere with the nuclear accumulation of mature SREBP. By showing that SREBP-1 has a key role in the resistance to mutant BRAF-targeted therapy our work identifies an important clinical setting in which such inhibitors may provide clear clinical benefit.

## Methods

**Cell culture**. A375 was obtained from ATCC. FLCM was generated from melanoma derived from $Braf^{CA}$, $Tyr::CreER$ and $Pten^{lox4-5}$ mice. M202, M207, and M233 were gifted by professor A. Ribas. 451 and 451lu R, M229, M229 R, M238, M238 R, M249, and M249 R were gifted by R. Lo. A101D BMR and D10 BMR were kindly gifted by Professor Daniel Peeper. NHEM was obtained from melanocytes derived from the foreskin of a pool of three healthy neonatal donors. The procedure was approved by the ethical committee of the University of Leuven and executed according to Helsinki guidelines.

All cell lines were propagated in DMEM High Glucose (Sigma), supplemented with 10% FBS (Gibco Lot 41F4234K) and 4 mM glutamine (ThermoFisher). A101D BMR and D10 BMR growth media was additionally supplemented with dabrafenib and trametinib. NHEM were cultured in Medium 254 (ThermoFisher) supplemented with HMGS (ThermoFisher) and Antibiotic-Antimycotic to 1×(ThermoFisher). 451lu R SREBF-1 KO clones were grown in DMEM High Glucose supplemented with 30% FBS and 4 mM glutamine. All cell cultures were periodically tested for mycoplasma contamination. All experiments were performed in DMEM High Glucose, supplemented with 2% FBS (Gibco Lot 41F4234K) and 4 mM glutamine, except for $^{13}$C-glucose metabolite tracer studies, where 4.5 g L$^{-1}$ $^{13}$C-glucose (Cambridge isotope laboratories) was supplemented to DMEM no glucose (ThermoFisher). The following compounds were used at the stated concentrations. NAC (120 μM) from Sigma, alpha-tocopherol (100 μM) from Sigma, ferrostatin (1.25 μM) from Sigma, vemurafenib (5 μM) from ApexBio, dabrafenib (2.5 μM) from Selleckchem, trametinib (0.5 μM) from Selleckchem, betulin (2 or 3 μM) from Sigma, fatostatin (0.5 or 1.5 μM) from Tocris Bioscience, oleic acid (20 μM) from Sigma, linoleic acid (10 μM) from Sigma and linolenic acid (10 μM) from Sigma.

**RNA-seq**. RNA concentration and purity were determined spectrophotometrically using a Nanodrop ND-1000 (Nanodrop Technologies) and RNA integrity was assessed using a Bioanalyser 2100 (Agilent). Samples were analyzed on an HiSeq2000 (Illumina). The raw sequencing data are publically available and can be accessed at https://www.ncbi.nlm.nih.gov/sra/SRP143504.

**Plasmid transfections**. M202 and M207 were transfected by electroporation (Neon Transfection System, ThermoFisher) with either pBabe-puro or pBabe-puro-BRAF$^{V600E}$. A375, 451lu, and M249 were transfected by electroporation (Neon Transfection System, ThermoFisher) with pbabe-puro-HA-SREBF-1 (Y335R)-myc.

**Construction of knockout cell lines**. 451lu R cells were transfected (Neon Transfection System, ThermoFisher) with CRISPR-Cas9 plasmid constructs with a guide-RNA targeting human SREBF-1 exon 1 (VectorU6gRNA-Cas9-2A-GFP, target ID: HS0000039707 and HS0000039709) (Sigma). 72 h post-transfection, the top 10% of GFP expressers were sorted by FACS (BD bioscience, ARIA III) into single wells for colony formation. Targeted regions of individual clones were

sequenced (Sanger sequencing, LGC) and indels in allelic sequences we genotyped using CRISP-ID (Supplementary Fig. 9c)[39].

**$^{14}$C-acetate incorporation into lipids and $^{13}$C-glucose tracing**. Cells were grown in 6-well plates up to 80% confluence and were treated with 0.1 μCi acetate-2-$^{14}$C (55 mCi/mmol; Amersham) for 4 h. After three washes with PBS (Sigma), cells were trypsinized and resuspended in 1 mL PBS, followed by sonication. 0.125 mL of lysate was set aside for DNA measurement and 0.7 mL of lysate was mixed with 0.9 mL MeOH solution (MeOH:HCl = 8:1) (Sigma) and 0.8 mL CHCl$_3$ (Sigma). After vortexing the mixture for 30 s followed by centrifugation for 10 min at $2000 \times g$, the organic fraction was counted for radioactivity (Tri-Carb 2810 TR scintillation counter, PerkinElmer). Mass spectrometry analysis of $^{13}$C-glucose incorporation into palmitate and subsequent isotopomer spectral analysis was performed as described previously[63].

**Untargeted metabolomics**. Cell extracts were separated on an Acquity HSS T3 UPLC column (Waters Corp, 2.1 mm × 150 mm, 1.8 μm particle size) using an Ultimate 3000 HPLC (Dionex ThermoFisher Scientific Inc). Elution of metabolites was performed using a quaternary solvent system. The data was collected over a mass range of 50–1050 m/z. Data analysis was done using ThermoFisher Scientific Quan software (Xcalibur version 4.0) and manually verified. Further analysis was done using in-house software tools.

**Cholesterol amount and uptake quantification**. Cholesterol was quantified using Amplex Red cholesterol assay kit (ThermoFisher). Cholesterol uptake was quantified using NBD cholesterol (ThermoFisher) according to ref. [64].

**Analysis of intact phospholipid species by ESI-MS/MS**. 0.7 mL of homogenized tissue or cells were mixed with 0.9 mL MeOH:HCl(1 N) (8:1), 0.8 mL CHCl$_3$ and 200 μg mL$^{-1}$ of the antioxidant 2,6-di-tert-butyl-4-methylphenol (Sigma). The organic fractions were evaporated under vacuum, reconstituted in MeOH/CHCl$_3$/NH$_4$OH (90:10:1.25) and lipid standards were added (Avanti Polar Lipids). Phospholipids were analyzed by electrospray ionization tandem mass spectrometry (ESI-MS/MS) on a hybrid quadrupole linear ion trap mass spectrometer (4000 QTRAP system, AB SCIEX) equipped with a TriVersa NanoMate robotic nano-source (Advion Biosciences) as described[26].

**Immunoblotting analysis**. Following ice-cold PBS washes, cells were collected in sample buffer (ThermoFisher) supplemented with DTT (Sigma), sonicated and boiled for 5 min. Equal amounts of protein were loaded onto precast gels (NuPAGE, ThermoFisher), transferred to nitrocellulose membranes, and incubated with antibodies against SREBP-1 (1/1000 dilution) (Active Motif, #39939), phospho-MEK1/2 ser217/221 (1/1000 dilution) (Cell Signaling, #9154); GAPDH (1/20000 dilution) (Cell Signaling, #5174), myc-Tag (1/2000 dilution) (Cell Signaling, #2276), HA-Tag (1/5000 dilution) (Cell Signaling, #3724), phosphor-AKT ser473 (1/1000 dilution) (ThermoFischer, #98H9L8), pS6 ser235 (1/1000 dilution) (Cell Signaling, #2211), and pERK 1/2 (1/1000 dilution) (Cell Signaling, #9101). Full unedited blots are shown in Supplementary Data 1.

**RNA extraction and RT-qPCR**. RNA extraction and RT-qPCR were performed as described previously[24]. Primers used were: FASN: Fw 5′-TCCGAGATTCCATC CTACGC-3′, Rv 5′-GCAGCTGTGACACCTTCAGG-3′, ACLY: Fw 5′-TGTAA CAGAGCCAGGAACCC-3′, Rv 5′-CTGTACCCCAGTGGCTGTTT-3′, ACACA: Fw 5′-TGAACTTCACACAGGTAGTCTGCC-3′, Rv 5′-TGGAACACTCGATGG AGTTTCT-3′, SREBP-1a: Fw 5′-GCTGCTGACCGACATCGAA-3′, SREBP-1c: Fw 5′-GGAGCCATGGATTGCACTTT-3′, SREBP-1a/1c Rv 5′- TCAAATAGGCCAG GGAAGTCA-3′, and 18 S as a reference gene: Fw 5′-CGCCGCTAGAGGTGA AATTC-3′, Rv 5′-TTGGCAAATGCTTTCGCTC -3′.

**Confocal microscopy**. Cells were transfected (Neon Transfection System, ThermoFisher) with a plasmid coding an HA and myc-tagged transcriptionally inactive SREBP-1 (Y335R) (GenScript). The SREBP cDNA was obtained from pTK-HSV-BP1a[65]. Cells were treated as indicated, fixed and incubated with antibodies against: myc-Tag (1/2000 dilution) (Cell Signaling, #2276) and HA-Tag (1/5000 dilution) (Cell Signaling, #3724), GM130 (1/500 dilution) (BD Biosciences, #610822), and PDI (1/1000 dilution) (ThermoFisher, #MA3-018). Samples were imaged using an Olympus FluoView FV1000 confocal microscope.

**Proliferation assays**. Proliferation curves were generated using an IncuCyte ZOOM system (Essen BioScience) on cells seeded of microplates (TPP) based on phase contrast images taken at 2 h intervals for the duration of the experiments. Colony formation in soft agar was performed as described by Franken et al.[66] Except, colonies were stained with Vybrant DyeCycle Green nuclear stain (ThermoFisher), imaged with a Typhoon FLA 9500 laser scanner (GE Healthcare) and quantified in ImageJ.

**Mitochondrial ROS measurement**. Mitochondrial ROS was measured using MitoSOX red mitochondrial superoxide indicator (ThermoFisher) according to the manufacturer's instructions. Cells were assayed using a FACS Verse flow cytometer (BD Biosciences).

**Lipid peroxidation assay**. Lipid peroxidation was quantified using the MDA assay kit (Sigma) according to manufacturer's instructions with some exceptions. Briefly, cells or tissue were collected in BHT supplemented PBS. TBA-acetic acid solution was buffered to pH 3.5. Plates were read using an EnSpire Multimode Plate Reader (PerkinElmer).

**Animal experiments**. The Mel006 PDX model was derived from an in-transit metastasis from a patient undergoing surgery as part of standard-of-care melanoma treatment at the University Hospitals Leuven. The patient provided written informed consent and this procedure was approved by the UZ Leuven Medical Ethical Committee (S54185) and carried out in accordance with the principles of the Declaration of Helsinki. All procedures involving animals were carried out in accordance with the guidelines of the IACUC and the Animal Care and Use Ethical Committee (KU Leuven, P038/2015). Fresh tumor tissue carrying the BRAF$^{V600E}$ mutation was collected in RPMI640 medium with antibiotics, rinsed in PBS and transplanted subcutaneously in the interscapular fat pad of female SCID-beige mice (Taconic). Sedation and analgesia was performed using ketamine, medetomidine and buprenorphine. Upon reaching generation 3, tumor fragments were transplanted into nude mice (NMRI-Fox1nu, Taconic) and when tumor size reached 200 mm$^3$, mice were randomly assigned to a cohort and drugs or vehicles were blindly administered daily by oral gavage. Vemurafenib and fatostatin were both administered daily at 20 mg/kg. fatostatin or vehicle (30% PEG (Sigma) in water (Baxter)) was administered an hour after vemurafenib or vehicle (2% hydroxypropylcellulose (Sigma) in water (Baxter)). Tumor size was measured blindly with digital calipers (Fowler Sylvac) every 3 days. Mice were sacrificed at 28 days following start of treatment or when tumors reached a volume of 1500 mm$^3$. The investigators were blinded for the evaluation of the results.

**Statistical analysis**. The results were analyzed in GraphPad Prism 6.0 h using a *t*-test. In case of multiple comparisons a correction was applied using the Holm–Sidak method. *p*-values of <0.05 were considered to be statistically significant. All data presented represent means ± s.e.m.

**Data availability**. The authors declare that the data supporting the findings of this study are available within the paper and its Supplementary Information files.

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

## Acknowledgements

J.D. and A.T. are recipients of a research fellowship from the Flemish Agency for Innovation by Science and Technology (IWT). F.B. is funded by MSCA-ITN-2014 Horizon 2020 project MELPLEX. J.A.G.D. is funded by a FWO postdoc fellowship. This work was supported by grants C16/15/073 and C32/17/052 from the KU Leuven, Interreg V-A EMR23 EURLIPIDS, G0E0817N from the Research Fund – Flanders (FWO) and a grant from the Belgian Foundation Against Cancer. S.-M.F. acknowledges funding support from Marie Curie – CIG, FWO – Odysseus II, FWO – Research Grants, Eugène Yourassowsky Schenking, KU Leuven – Methusalem Co-Funding, and Bayer Health Care (grants4targets). We would like to thank Dr. Flavie Luciani for the kind gift of FLCM, Professor R. Lo for the kind gift of 451lu, 451lu R, M229, M229 R, M238, M238 R, M249, and M249 R and Professor A. Ribas for the kind gift of M202 and M207, and Professor Daniel Peeper for the kind gift of A101D BMR and D10 BMR. FACS was performed by the KU Leuven FACS Core Facility, for which we would like to thank Ms. Pier Andrée Penttila for expert technical assistance, and RNA-Seq was performed by the VIB Nucleomics Core (www.nucleomics.be) for which we would like to thank Dr. Rudy van Eijsden for expert technical assistance. We would like to thank the KU Leuven PDX platform, TRACE (www.uzleuven-kuleuven.be/lki/trace), especially Frédéric Amant, Els Hermans, Debby Tomas, and Ellen Gommé for expert assistance with the in vivo experiments. We would like to thank Prof Dr Bart Ghesquière and the Metabolomics Expertise Center from the department of Oncology (KU Leuven) and the Center for Cancer Biology (CCB, VIB) for the metabolomics analyses. We would like to thank Mr. Jens Wouters for expert technical assistance. We would like to thank Professor Sebastian Munck for expert guidance with ICC. We would like to thank the Goldstein and Brown lab for the kind gift of pTK-HSV-BP1a.

## Author contributions

A.T., J.D., and J.V.S. designed the experiments and wrote the paper. A.T., J.D., and F.V. performed the experiments. J.-C.M. provided extensive helpful comments and discussion. A.T. and J.D. analyzed data. J.V.S. oversaw the project and acquired funding. F.R. analyzed and generated the TCGA, RNA-seq data. A.R. established and characterized the Mel006 PDX model. R.D. executed the phospholipidomics analysis. E.W. allowed use of the mass spectrometer. D.N., K.V.d.E., F.B., and J.V.d.O. generated the mouse IHC slides. K.N., M.V.P., and P.A. extracted the NHEM. J.A.G.D. and S.-M.F. performed and analyzed the stable isotope metabolic flux experiments.

## Additional information

**Competing interests:** The authors declare no competing interests.

