## [Peer Review File · Nature Communications]

Reviewers' comments:

Reviewer #1 (Remarks to the Author):

This manuscript demonstrates that induction of lipid biosynthesis contributes to therapy resistance in B-Raf mutant melanoma. The authors have analysed gene expression data from vemurafenib treated cells and found that enzymes involved in fatty acid metabolism are not downregulated in treatment resistant cells. They also show that resistant cells fail to downregulate de novo lipid synthesis in response to vemurafenib treatment. Lipidomic analysis showed characteristic increase in poly-unsaturated fatty acid species indicative of decreased de novo synthesis with compensatory increase in poly-unsaturated species. They go on to show that vemurafenib blocks SREBP-1 activation only in sensitive cells and that inhibition of SREBP-1 causes the same changes to lipid composition in resistant cells as seen for vemurafenib in sensitive cells, i.e. decrease of mono-unsaturated and a compensatory increase in poly-unsaturated lipid species. Inhibition of SREBP-1 by the SCAP inhibitors fatostatin or botulin caused re-sensitisation of resistant cells to vemurafenib treatment, potentially by increasing lipid peroxidation. This was supported by the observation that a mono-unsaturated fatty acid, oleic acid, could restore proliferation in vemurafenib resistant cells after SREBP-1 inhibition, while addition of poly-unsaturated fatty acids, linoleate and linolenate, exacerbated growth inhibition. Moreover, the authors could detect evidence of lipid peroxidation in resistant cells after combined vemurafenib and fatostatin treatment. This was reduced by oleic acid but enhanced by poly-unsaturated fatty acids. Finally, the authors show that fatostatin restores sensitivity to vemurafenib in resistant PDX tumours, which was accompanied by a shift to more poly-unsaturated fatty acid species and increased lipid peroxidation. The authors conclude that SREBP-1 contributes to vemurafenib resistance in B-Raf mutant melanoma by maintaining levels of saturated and mono-unsaturated fatty acid species in cancer cells thereby preventing excess lipid peroxidation. Targeting SREBP-1 re-sensitises cells to the inhibitor and reduces tumour growth.

Overall, the manuscript makes a convincing case for the importance of SREBP-1 mediated lipid synthesis in vemurafenib resistance. This is particularly interesting in the context of the high incidence of therapy resistance found in melanoma and the findings of this study could have a direct impact on treatment strategies in this disease.

The experiments are generally of a very high standard and the results are conclusive and compelling. One point that could be explored deeper is the role of lipid peroxidation in restoring vemurafenib resistance in B-Raf mutant cells. The authors use MDA detection to determine lipid peroxidation. While this assay is certainly useful, it only detects a by-product of lipid peroxidation. It should be considered to also include a more direct method to determine lipid peroxidation, either by using fluorescent dyes or by detecting peroxidised lipid species by mass spectrometry. The authors should also use lipid-specific anti-oxidants (e.g. alpha-tocopherol or ferostatin) rather than NAC to confirm that lipid peroxidation is indeed involved in the re-sensitisation. Moreover, they should include NAC treatment also in non-vemurafenib treated cells to exclude any major effects of this compound on cell proliferation (data shown in Figure 4D). In addition, the authors have not fully addressed the mechanism of induction of lipid peroxidation following combined inhibition of B-Raf and SREBP-1. In this context, it would be interesting to establish whether lipid peroxidation is also a major mode of vemurafenib toxicity in sensitive cells (which show the shift to poly-unsaturated lipids)? Why do the authors only see a minor increase in lipid peroxidation after fatostatin treatment alone even though there is already a major shift in lipid composition? Or, alternatively, how do the authors explain the further increase in poly-unsaturated fatty acid species by vemurafenib in cells treated with SREBP-1 inhibitors? The fact that vemurafenib alone already induces a substantial increase in MDA levels in resistant cells (data shown in Figure 4C) points to a more direct mechanism. The authors should aim to address these questions to further increase the impact of their findings.

Apart from this criticism, there are a few minor points that could be addressed.

Specific comments:

Abstract: Avoid "well" after vulnerabilities. Repetition of "may offer".

Page 5: "Interestingly" is used in two consecutive sentences. Replace "though" with "through" and rephrase the next sentence.

Page 8: Exposure of.. should this be 45lu R cells?

There are also some problems with commas and consistency of the figure labelling.

P-value indicators are missing from legend of Figure 4.

Reviewer #2 (Remarks to the Author):

In this manuscript, the authors provide a series of experiments pointing to the importance of down-regulation of SREBP1 which leads to lipogenesis and escape from BRAF inhibition in BRAF V600E-mutated melanomas.

1. The authors use a variety of different BRAF inhibitor resistant cell lines, with diverse mechanisms of resistance, to show that the lipogenesis effects supervene over the specific mechanism of resistance. To what extent does the specific pathway activated in the resistant cells impact upon SREBP1 expression or downstream genes. For instance, in cells with activated PI3K as a form of resistance, how do PI3K inhibitors affect SREBP1? It is possible that downstream central mediators such as ERK or pS6 are important controllers of the SREBP1 response.

2. Along the same lines, do the authors have any information about how lipogenesis is altered in cells that are resistant to dual BRAF/MEK or triple BRAF/MEK/ERK inhibition? This is potentially important since it is unusual to use BRAF monotherapy, and instead received combined MAPK inhibition. This information would be of translational potential that would be of interest to the community.

3. Is there a direct transcriptional effect of BRAFV600E to increase SREBP1? The proposed mechanism is that BRAF inhibition affects proteosomal degradation of SREBP1, but this does not necessarily show a direct connection between BRAF (or its downstream activators) and SREBP1.

4. The lipid effects are primarily centered upon the role of phospholipids as the major end product of lipogenesis from BRAF. However, another possibility is that some of these lipids are being shunted to be used for beta-oxidation and ATP synthesis. Have you been able to measure this through Seahorse type assays, or do you think the primary effect is solely on membrane phospholipids.

5. The findings regarding lipid peroxidation after BRAF inhibitor resistance are an interesting mechanism. Do you have data on how MDA levels change during the development of BRAF inhibitor resistance, i.e. is this changed early in the course or once resistance is established? Do you envision the peroxidation effect as a side effect of resistance or a primary driver of resistance?

6. For the tumor xenograft studies, do the combinations also lead to apoptosis or senescence of the tumor cells, since a decrease in Ki67 is only one mechanism of reducing overall tumor growth. If there is a population of senescent cells left behind after combined treatment, this would raise concerns that these cells could later emerge as resistant to both vemurafenib and falostatin down the road.

7. Figure S1B: Why aren't baseline levels of lipogenic enzymes higher in the resistant clones?

8. Figure S2: The effect on de novo palmitate synthesis is not consistently seen in all cell lines. This should be pointed out.

9. Figure 1E: The difference between the sensitive and resistant lines in the shift in lipid saturation

is fairly subtle from this heat map.

10. In general, the Western blots for mSREBP-1 are of poor quality limiting the ability of the reader to be convinced. Some of the figures, such as Fig 2E and S13, are not interpretable.

11. Pg 7, 1st paragraph "The active form...": This is a distraction, the blots are terrible, and the paragraph could be deleted.

12. Pg 9: They describe the effect of vem + SREBP-1 inhibitors on the resistant cell lines shown in Fig 3C as "striking". The effect shown is, in fact, quite modest. The effect of vem on reducing growth of the heterozygote knock out (Fig 3F) is described as "strong" whereas the effect is, again, modest. The authors should avoid descriptors that oversell the data.

13. Figure 4: A) the effects shown here are quite modest and of questionable clinical significance. B) Some of the effects here are also quite small. They point out the effect of adding oleate to Vem/fatostatin which is statistically different but it is hard to believe this difference is biologically meaningful.

14. Figure 5: They should explain why vemurafenib treatment alone made such a small difference on tumor growth.

Minor points

Introduction: Statement that "nearly all cases" develop resistance to BRAFi is not exactly true. The relapse rate is about 80%. They should be more precise and could give references (e.g. Long et al. Lancet Oncol 17:1743, 2016 is one example).

There are a few sentences that seem to be missing words. Pg 7 at the bottom ("In contrast,...") and pg 10 towards the top ("The above data...") are examples.

Figures S6 and S8 contain photographs for cell culture confluency which are difficult to see on line and will probably be impossible in print. They should include a quantification of confluency in these figures.

Reply to the reviewers' comments

We thank the editor and the reviewers for their critical and constructive comments, which we address here point by point. We feel that these comments have substantially improved our manuscript and are grateful for this opportunity.

Reviewer #1:

This manuscript demonstrates that induction of lipid biosynthesis contributes to therapy resistance in B-Raf mutant melanoma. The authors have analysed gene expression data from vemurafenib treated cells and found that enzymes involved in fatty acid metabolism are not downregulated in treatment resistant cells. They also show that resistant cells fail to downregulate de novo lipid synthesis in response to vemurafenib treatment. Lipidomic analysis showed characteristic increase in poly-unsaturated fatty acid species indicative of decreased de novo synthesis with compensatory increase in poly-unsaturated species. They go on to show that vemurafenib blocks SREBP-1 activation only in sensitive cells and that inhibition of SREBP-1 causes the same changes to lipid composition in resistant cells as seen for vemurafenib in sensitive cells, i.e. decrease of mono-unsaturated and a compensatory increase in poly-unsaturated lipid species. Inhibition of SREBP-1 by the SCAP inhibitors fatostatin or botulin caused re-sensitisation of resistant cells to vemurafenib treatment, potentially by increasing lipid peroxidation. This was supported by the observation that a mono-unsaturated fatty acid, oleic acid, could restore proliferation in vemurafenib resistant cells after SREBP-1 inhibition, while addition of poly-unsaturated fatty acids, linoleate and linolenate, exacerbated growth inhibition. Moreover, the authors could detect evidence of lipid peroxidation in resistant cells after combined vemurafenib and fatostatin treatment. This was reduced by oleic acid but enhanced by poly-unsaturated fatty acids. Finally, the authors show that fatostatin restores sensitivity to vemurafenib in resistant PDX tumours, which was accompanied by a shift to more poly-unsaturated fatty acid species and increased lipid peroxidation. The authors conclude that SREBP-1 contributes to vemurafenib resistance in B-Raf mutant melanoma by maintaining levels of saturated and mono-unsaturated fatty acid species in cancer cells thereby preventing excess lipid peroxidation. Targeting SREBP-1 re-sensitises cells to the inhibitor and reduces tumour growth.

Overall, the manuscript makes a convincing case for the importance of SREBP-1 mediated lipid synthesis in vemurafenib resistance. This is particularly interesting in the context of the high incidence of therapy resistance found in melanoma and the findings of this study could have a direct impact on treatment strategies in this disease. The experiments are generally of a very high standard and the results are conclusive and compelling.

We thank this referee for his/her careful reading of our manuscript, the overall positive assessment of the work and his/her constructive comments.

One point that could be explored deeper is the role of lipid peroxidation in restoring vemurafenib resistance in B-Raf mutant cells. The authors use MDA detection to determine lipid peroxidation. While this assay is certainly useful, it only detects a by-product of lipid peroxidation. It should be considered to also include a more direct method to determine lipid peroxidation, either by using fluorescent dyes or by detecting peroxidised lipid species by mass spectrometry. The authors should also use lipid-specific anti-oxidants (e.g. alpha-tocopherol or ferostatin) rather than NAC to confirm that lipid peroxidation is indeed involved in the re-sensitisation. Moreover, they should include NAC treatment also in non-vemurafenib treated cells to exclude any major effects of this compound on cell proliferation (data shown in Figure 4D).

We thank the reviewer for this comment. As suggested, we have attempted to use additional, more direct methods to measure lipid peroxidation. We have tried using BODIPY dyes as recommended by this referee, the commercial Click It lipid peroxidation kit from Life Technologies and a ferrous ion based lipid peroxidation kit. Unfortunately, none of these assays were sensitive enough to detect the kind of changes in lipid peroxidation that occur in the physiological context of our experimental system. Note that also some of these assays (e.g. the click it lipid peroxidation kit) actually measure the degradation of specific lipids, and are therefore also not measuring lipid peroxidation directly.

Direct measurement of peroxidized lipids by mass spectrometry (MS) would be an obvious alternative. One of the challenges associated with MS-based techniques for quantitative detection of peroxidized lipids is, however, that there are myriad of oxidized lipid species, which all exist in minute amounts. Detecting these requires advanced protocols and sensitive instruments. To date, only a handful of highly specialized laboratories can perform such experiments. Our MS/MS triple quadrupole is certainly not suitable for this type of analysis and, unfortunately, we have not been able to set up such an experiment in collaboration during the course of the revision process.

We would like to argue, however, that the MDA assay that we have used, although providing an indirect measure, is a very sensitive and well-suited and accepted method to, at least partly, address this question.

Importantly, to further confirm the contribution of lipid peroxidation in restoring vemurafenib resistance, as suggested by this referee, we compared the effects of the antioxidant NAC with other lipid-specific anti-oxidants such as alpha-tocopherol or ferostatin. We now show the effects of the antioxidants NAC, ferostatin and alpha-tocopherol in combination with control, fatostatin, vemurafenib and vemurafenib + fatostatin conditions. The new data indicate that combining these antioxidants rescues cell proliferation. Interestingly, the lipid soluble

antioxidants led to an even greater proliferation rescue than NAC. These new data are shown in revised Figure 4D. Also in other new panels (e.g. Figure 4E) we have included rescue experiments with lipid-specific antioxidants. We feel that these new data and this combination of approaches, including MDA measurements and rescue by lipid-specific antioxidants, provide sufficient evidence for the involvement of lipid peroxidation.

Moreover, we have included NAC treatment also in non-vemurafenib treated cells, as requested by the reviewer. The first panel of Figure 4D now confirms that this compound alone does not have major effects on cell proliferation.

Figure 4D

Effect of alpha-tocopherol, ferrostatin and NAC supplementation on cell proliferation of 451lu R cells treated with vemurafenib + fatostatin.

In addition, the authors have not fully addressed the mechanism of induction of lipid peroxidation following combined inhibition of B-Raf and SREBP-1. In this context, it would be interesting to establish whether lipid peroxidation is also a major mode of vemurafenib toxicity in sensitive cells (which show the shift to poly-unsaturated lipids)?

This is indeed a very important point and we thank the referee for making this comment.

In order to address this point, we performed additional experiments. We co-treated the drug sensitive cell lines M229 and 451lu with a very low dose of vemurafenib (150 nMol) and SREBP inhibitors. The combined inhibition enhanced the cytostatic activity of vemurafenib. Importantly, the cytostatic activity of vemurafenib was, at least partly, rescued by lipid-specific antioxidants, indicating that lipid peroxidation indeed contributes to vemurafenib cytotoxicity. These data have been included in figure 4E and Supplementary Figure S24 of the revised manuscript.

Figure 4E

Proliferation curves of the vemurafenib responsive cell lines M229 and 451lu treated with vemurafenib, alpha-tocopherol, ferrostatin and NAC (phase contrast density measured by the Incucyte system).

Figure S24

Proliferation curves of the vemurafenib responsive cell lines M229 and 451lu treated with vemurafenib, betulin and fatostatin (phase contrast density measured by the Incucyte system).

Why do the authors only see a minor increase in lipid peroxidation after fatostatin treatment alone even though there is already a major shift in lipid composition? Or, alternatively, how do the authors explain the further increase in poly-unsaturated fatty acid species by vemurafenib in cells treated with SREBP-1 inhibitors? The fact that vemurafenib alone already induces a substantial increase in MDA levels in resistant cells (data shown in Figure 4C) points to a more direct mechanism. The authors should aim to address these questions to further increase the impact of their findings.

Lipid peroxidation requires the presence of poly-unsaturated lipids and ROS. Fatostatin treatment increases the degree of polyunsaturation but does not significantly increase ROS. Hence, it increases the cell's susceptibility to lipid peroxidation by other triggers of ROS, but has only modest effects on lipid peroxidation per se under low ROS stress conditions. Vemurafenib itself is a well-known inducer of ROS through various mechanisms^{1,2}. It may cause membrane lipid peroxidation to an extent, even in resistant cells, which is further amplified through increasing the ratio of polyunsaturated lipids, e.g. by SREBP-1 inhibition or supplementation with exogenous polyunsaturated fatty acids.

In several of the models used, inhibition of SREBP-1 (by vemurafenib or fatostatin) is partial, explaining additive effects. In the full genetic KO of SREBP-1 in figure 3B, vemurafenib exposure did not further increase the amount of poly-unsaturated lipid species, suggesting that these changes to membrane lipid polyunsaturation occur through SREBP-1.

Specific comments:

Abstract: Avoid "well" after vulnerabilities. Repetition of "may offer".

Page 5: "Interestingly" is used in two consecutive sentences. Replace "though" with "through" and rephrase the next sentence.

Page 8: Exposure of.. should this be 45lu R cells?

There are also some problems with commas and consistency of the figure labelling.

P-value indicators are missing from legend of Figure 4.

We thank the referee for all of these comments/remarks, all of which we took into account to prepare the enclosed revised version of our manuscript.

Reviewer #2 (Remarks to the Author):

In this manuscript, the authors provide a series of experiments pointing to the importance of down-regulation of SREBP1 which leads to lipogenesis and escape from BRAF inhibition in BRAF V600E-mutated melanomas.

1. The authors use a variety of different BRAF inhibitor resistant cell lines, with diverse mechanisms of resistance, to show that the lipogenesis effects supervene over the specific mechanism of resistance. To what extent does the specific pathway activated in the resistant cells impact upon SREBP1 expression or downstream genes. For instance, in cells with activated PI3K as a form of resistance, how do PI3K inhibitors affect SREBP1? It is possible that downstream central mediators such as ERK or pS6 are important controllers of the SREBP1 response.

We thank the reviewer for this comment. It is well-known that pathways such as PI3K and MEK-ERK affect SREBP³⁻⁵. It is therefore indeed quite plausible that these pathways, which are often (re)activated in resistant cells, contribute to sustained activation of SREBP in therapy-resistant cells. To test this possibility, we assessed levels and activity of SREBP in resistant cells upon treatment with BRAF, PI3K/Akt and Mek inhibitors. Unexpectedly, the consequences of AKT inactivation, which is lethal in some of the resistant cell lines tested (not shown), did not lead to a reproducible/consistent effect on SREBP1 levels.

Interestingly, however, we have observed a consistent role for MEK-ERK pathway activation in SREBP1 processing and transcriptional activity. Indeed, MEK inhibition as a result of Trametinib exposure caused a severe decrease in mature SREBP1 (mSREBP1) protein levels and expression of a series of its well-established target genes. As an example of this we show below the effects of MEK-inhibition in 451lu R cells (Supplementary figures S7-8). These new data –establishing a key role for ERK-MEK re-activation in SREBP1 processing- have been included in the revised version of the manuscript.

Figure S7 Western blot analysis of pERK, GAPDH and SREBP-1 in the cell lines 451lu and 451luR in response to vemurafenib (5 μ M), dabrafenib (2.5 μ M) and trametinib (0.5 μ M).

Figure S8 QPCR analysis of ACLY, FASN and ACACA normalized to 18S, in 451luR in response to vemurafenib, dabrafenib or trametinib.

2. Along the same lines, do the authors have any information about how lipogenesis is altered in cells that are resistant to dual BRAF/MEK or triple BRAF/MEK/ERK inhibition? This is potentially important since it is unusual to use BRAF monotherapy, and instead received combined MAPK inhibition. This information would be of translational potential that would

be of interest to the community.

We thank the reviewer for this important comment. Dual RAF/MEK inhibition is indeed a standard of care for BRAF-mutant melanoma patients and although this treatment leads to longer-lasting anti-tumor responses in patients compared to single RAF-inhibition, the vast majority of patients do eventually relapse. We therefore treated two dual resistant cell lines (A101D BMR and D10 BMR) with Dabrafenib + Trametinib in combination with Betulin and Fatostatin. Importantly, betulin and fatostatin treatment further sensitized these cells to dual RAF/MEK-inhibition. These important data have been included in the revised manuscript (Supplementary figures S17-20).

Figure S17

Proliferation curves of the dabrafenib + trametinib double resistant cell line A101D BMR treated with dabrafenib, trametinib, betulin and fatostatin (phase contrast density measured by the Incucyte system).

Figure S19

Proliferation curves of the dabrafenib + trametinib double resistant cell line D10 BMR treated with dabrafenib, trametinib, betulin and fatostatin (phase contrast density measured by the Incucyte system).

3. Is there a direct transcriptional effect of BRAFV600E to increase SREBP1? The proposed mechanism is that BRAF inhibition affects proteasomal degradation of SREBP1, but this does not necessarily show a direct connection between BRAF (or its downstream activators) and SREBP1.

To address this important question we assessed SREBP1 mRNA levels in various cell lines exposed to the BRAF-inhibitor. We could not see any significant differences in most cells. In 451lu and A375, however, there was a slight but reproducible decrease in both SREBP1a and SREBP1c transcript levels upon BRAF inhibition in the sensitive but not isogenic resistant lines.

We have added the qPCR data on SREBP mRNA expression in supplementary Figure S4. We conclude that most of the observed effects are mediated by post-transcriptional mechanism(s), although we cannot rule out that in some specific cellular context modulation of SREBP1 transcription may also account for these effects, especially as SREBPs auto-regulate their expression.

Figure S4 The full cell line panel was subjected to qPCR for SREBP1a and SREBP1c after Vemurafenib (5 μ M) treatment, data are represented as mean \pm SEM.

4. The lipid effects are primarily centered upon the role of phospholipids as the major end product of lipogenesis from BRAF. However, another possibility is that some of these lipids are being shunted to be used for beta-oxidation and ATP synthesis. Have you been able to measure this through Seahorse type assays, or do you think the primary effect is solely on membrane phospholipids.

We would like to thank the reviewer for this comment. To address this, we performed untargeted metabolomics experiments and did not see a major difference in ATP levels, nor energy charge.

Interestingly, we found that under combined BRAF and SREBP inhibition, melanoma cells have markedly reduced antioxidant potential as indicated by the NAD/NADH, NADP/NADPH and GSSG/GSH ratios, whereas energy potential were not affected. This is in line with our hypothesis that membrane lipid poly-unsaturation renders these cells susceptible to ROS-induced damage and lowers their antioxidant potential. These data have been included in the manuscript in Supplementary Figure S22. Together, these data corroborated our conclusion that membrane lipid profile and lipid peroxidation play important roles in these models, as opposed to ATP generation or energy charge alone.

Figure S22

Relative ATP concentration, energy charge, NAD/NADH, NADP/NADPH and GSSG/GSH ratios in 451luR were obtained by untargeted metabolomics following vemurafenib and fatostatin treatment.

5. The findings regarding lipid peroxidation after BRAF inhibitor resistance are an interesting mechanism. Do you have data on how MDA levels change during the development of BRAF inhibitor resistance, i.e. is this changed early in the course or once resistance is established? Do you envision the peroxidation effect as a side effect of resistance or a primary driver of resistance?

This is an interesting point indeed. In order to address this, we performed histopathological analyses of available tumor sections from the Mel006 PDX model that were either left untreated (drug-naïve) or exposed to dabrafenib and trametinib for a few days (early drug response phase), for 2 weeks (persister state) and for a period of more than 2 months (acquired resistance) (please note that the TBARS/MDA assay must be performed on fresh frozen or fresh samples and these were not available).

Whereas we observed an increase in MDA staining during the drug response phase, Persister (or drug-tolerant) cells, as well as the drug-resistant cells (which acquired the ability to grow in the presence of the drugs) exhibited lower MDA levels than the drug-naïve cells.

We show the result of this analysis in Supplementary figure S27 of the revised version of the manuscript (see also below). These data are in keeping with the model proposed in this manuscript and provide further (pre)clinical relevance to our findings.

It is therefore conceivable that downregulation of lipid peroxidation either through modulating membrane prolife and/or increasing the cell antioxidant potential contributes to resistance to BRAF/MEK inhibitors.

Figure S27

(A). Anti MDA quantified histological staining in mel06 tumors. Tumors were stratified into four groups, either the naïve tumors, early treatment following 2 – 4 days of mouse treatment, persister state following two to three weeks of treatment and the tumor resistant state where the tumor develops therapy resistance. Histological staining reveals a non-significant increase in tumor MDA staining shortly after treatment, which is reduced in the persister state and in the resistant tumor.

(B) Representative images of anti-MDA (red) staining, nuclei were counterstained with DAPI. Trend is not significant.

6. For the tumor xenograft studies, do the combinations also lead to apoptosis or senescence of the tumor cells, since a decrease in Ki67 is only one mechanism of reducing overall tumor growth. If there is a population of senescent cells left behind after combined treatment, this would raise concerns that these cells could later emerge as resistant to both vemurafenib and falostatin down the road.

This is certainly a very valid point. We did observe tumour shrinkage and therefore believe that cell death is likely involved in the tumor regression process. In the figure below, we show cleaved caspase 3 staining in various PDX lesions exposed to the indicated drug(s).

Unexpectedly, we did not observe a significant increase in cleaved caspase staining in lesions exposed to Fatostatin and vemurafenib. We cannot exclude that cells die through another PCD mechanism such as ferroptosis.

We have also assessed whether these treatments induced senescence; however, we could not detect beta-galactosidase staining in any of the experimental conditions.

Cleaved caspase 3 staining in PDX tumors that represent the median of their respective cohort.

7. Figure S1B: Why aren't baseline levels of lipogenic enzymes higher in the resistant clones?

We certainly do not want to claim that resistant cells have enhanced lipogenesis per se. We argue, instead, that the down-regulation of lipogenesis in resistant cells is either less pronounced or simply does not occur in response to BRAF inhibition. We therefore propose that sustained lipogenesis under BRAF-inhibition is what causes resistance.

We have now included the following sentence in the manuscript to highlight this:

“We conclude that lipogenesis is sustained in therapy-resistant cells when compared to therapy-sensitive cells upon vemurafenib treatment.”

8. Figure S2: The effect on de novo palmitate synthesis is not consistently seen in all cell lines. This should be pointed out.

The palmitate synthesis rates are in fact reduced upon vemurafenib treatment in all sensitive lines tested and this to a greater extent than in the vemurafenib-resistant lines. This is clear when looking specifically at fatty acid biosynthesis, as opposed to total lipogenesis.

We apologize for not making this clear in the previous version of the manuscript. In order to avoid the confusion, we have explicitly stated this in the revised manuscript. We thank the referee for this comment.

9. Figure 1E: The difference between the sensitive and resistant lines in the shift in lipid saturation is fairly subtle from this heat map.

We again thank the referee for this comment. To address this, we have included a polyunsaturation index graph in the revised version. This graph shows species by number of unsaturations per lipid (Figure 1F).

10. In general, the Western blots for mSREBP-1 are of poor quality limiting the ability of the reader to be convinced. Some of the figures, such as Fig 2E and S13, are not interpretable.

We would like to point out that detecting mature SREBP1 by Western blotting has traditionally been challenging (see also the original Goldstein and Brown papers, and all subsequently published work on this protein). This can be explained by the fact that the mature protein often consists of multiple forms, as it undergoes multiple forms of post translational modifications including hyper-phosphorylation, and only represents a small fraction of the total pool of SREBP1. We nevertheless repeated some of the blots in order to increase their quality/readability; these new blots are now presented in the revised version of the manuscript.

11. Pg 7, 1st paragraph “The active form...”: This is a distraction, the blots are terrible, and the paragraph could be deleted.

Our apologies for the poor quality of the blots. We have removed this panel as suggested.

12. Pg 9: They describe the effect of vem + SREBP-1 inhibitors on the resistant cell lines shown in Fig 3C as “striking”. The effect shown is, in fact, quite modest. The effect of vem on reducing growth of the heterozygote knock out (Fig 3F) is described as “strong” whereas the effect is, again, modest. The authors should avoid descriptors that oversell the data.

We adapted the text so that the description of the data better fits the observed effects.

13. Figure 4: A) the effects shown here are quite modest and of questionable clinical significance. B) Some of the effects here are also quite small. They point out the effect of adding oleate to Vem/fatostatin which is statistically different but it is hard to believe this difference is biologically meaningful.

We apologize for the confusion. The aim here is to simply show that the combined effects of SREBP and BRAF inhibition can be altered by supplementation with saturated or poly-unsaturated lipids. This intention is to show that this effect occurs though lipid profile changes. A mono-unsaturated fatty acid, oleic acid, could restore proliferation in vemurafenib resistant cells after SREBP-1 inhibition, while addition of poly-unsaturated fatty acids, linoleate and linolenate, exacerbated growth inhibition. This is purely a mechanistic supplement to show that the effects that we observed occurred though changes in lipid profile and can, in part, be modulated by affecting the lipid profile.

We certainly do not want to make claims about the clinical potential of this observation, but use the data as a mean to gain mechanistic insights.

14. Figure 5: They should explain why vemurafenib treatment alone made such a small difference on tumor growth.

This is indeed a very good point. The Chris Marine lab, our collaborator on this project, established a series of PDX melanoma models and has so far treated 6 different BRAF-mutant models with BRAFV600E-inhibitor alone. Although model-to-model variability in the response is observed, in none of these cases – including MEL006 - such a treatment has led to tumor regression. In most cases, this treatment leads to stabilization of the disease, at best. This is also what has been observed in other melanoma labs such as the lab of Daniel Peeper or Mark Schackleton. This can be explained by the fact that these mice are immunocompromised and that tumor regression often needs active recruitment of functional immune cells. For the purpose of our experiments we sought to use a model in which the response to BRAF-inhibition alone is modest (poor responder/intrinsic resistance). We therefore chose MEL006 as the model of choice; as expected (seen in previous experiments) exposure to vemurafenib alone only led to a modest inhibition of growth.

Minor points

Introduction: Statement that “nearly all cases” develop resistance to BRAFi is not exactly true. The relapse rate is about 80%. They should be more precise and could give references (e.g. Long et al. Lancet Oncol 17:1743, 2016 is one example).

There are a few sentences that seem to be missing words. Pg 7 at the bottom (“In contrast,...”) and pg 10 towards the top (“The above data...”) are examples.

We have adapted the text accordingly. We thank the referee for pointing this out.

Figures S6 and S8 contain photographs for cell culture confluency which are difficult to see on line and will probably be impossible in print. They should include a quantification of confluency in these figures.

The cell morphology images are already a supplement to quantified data presented in the confluence curve figures throughout the text. For example, Supplementary Figure S10 are screengrabs of time points used to generate Figure 3C. These images are there to support these data and to additionally show the cell morphology, which is of interest to some experts in the field. The images are of high resolution and can be zoomed in to see individual cells.

References

1. Haq, R. *et al.* Oncogenic BRAF Regulates Oxidative Metabolism via PGC1 α and MITF. *Cancer Cell* **23**, 302–315 (2013).
2. Cesi, G., Walbrecht, G., Zimmer, A., Kreis, S. & Haan, C. ROS production induced by BRAF inhibitor treatment rewires metabolic processes affecting cell growth of melanoma cells. *Mol. Cancer* **16**, 102 (2017).
3. Porstmann, T. *et al.* SREBP Activity Is Regulated by mTORC1 and Contributes to Akt-Dependent Cell Growth. *Cell Metab.* **8**, 224–236 (2008).
4. Kotzka, J. *et al.* Sterol regulatory element binding proteins (SREBP)-1a and SREBP-2 are linked to the MAP-kinase cascade. *J. Lipid Res.* **41**, 99–108 (2000).
5. Roth, G. *et al.* MAP kinases Erk1/2 phosphorylate sterol regulatory element-binding protein (SREBP)-1a at serine 117 in vitro. *J. Biol. Chem.* **275**, 33302–33307 (2000).
6. Viswanathan, V. S. *et al.* Dependency of a therapy-resistant state of cancer cells on a lipid peroxidase pathway. *Nature* **547**, 453 (2017).

Reviewers' comments:

Reviewer #1 (Remarks to the Author):

The authors have addressed all comments raised by this reviewer.

Reviewer #2 (Remarks to the Author):

The authors have extensively responded to the initial review and the manuscript is certainly improved. I remained concerned that several of the experimental results show quite small effects and may not be biologically significant.

For example, they state that proliferation of therapy-resistant cells was strongly inhibited by vemurafenib and SREBP-1 (Line 215). But figure 3C shows rather modest effects.

Fig 3E: The effect of vemurafenib is not entirely convincing. Baseline growth of the knock outs is quite impaired. The effect of vemurafenib on knockout #2 seems minimal. For knock out #1, vemurafenib does seem to further impair the growth.

Fig S17-19: This is not particularly convincing. It appears that A101D cells are only partially resistant to dabrafenib/trametinb and there is very little effect of fatostatin. The D10 BMR cell line is much more resistant and shows a small effect of fatostatin but no real effect of betulin. Overall, it's hard to interpret these data.

Fig 4D: These expts claim to show that antioxidants partially rescue cell proliferation from combined SREBP-1/vemurafenib inhibition. This effect is quite minimal. Is there a way to express a p value?

Response to Reviewers' comments:

Reviewer #1 (Remarks to the Author):

The authors have addressed all comments raised by this reviewer.

We thank the referee for her/his support and positive assessment of our revised manuscript.

Reviewer #2

The authors have extensively responded to the initial review and the manuscript is certainly improved.

We thank the referee for her/his support and overall positive assessment of our revised manuscript.

I remained concerned that several of the experimental results show quite small effects and may not be biologically significant.

The reviewer's concerns are all related to the 2D confluence curves generated using the live cell imaging IncuCyte instrument. This instrument generates highly reproducible real-time kinetic data. We generated long-term cell density curves (up to 360 hours) in order to monitor growth until cultures reach confluency. This allows one to compare growth conditions at multiple time points. We now provide bar graphs highlighting time points and test for statistical significance at those selected time points so that differences between growth conditions can be better appreciated and significance highlighted. Please find below answer to the specific points raised by this referee.

For example, they state that proliferation of therapy-resistant cells was strongly inhibited by vemurafenib and SREBP-1 (Line 215). But figure 3C shows rather modest effects.

The bar graph shown below demonstrates that the additive effects of the combination treatments are statistically significant, both in the therapy-sensitive and resistant cells.

Figure S11 Proliferation of 451lu (time point 96h) and 451lu R (time point 168h) cells treated with vemurafenib (5 μM) and fatostatin or betulin. Data was taken from the experiment shown in main figure 3C. Time points chosen represent those where the fastest growing conditions started approaching confluence.

Fig 3E: The effect of vemurafenib is not entirely convincing. Baseline growth of the knock outs is quite impaired. The effect of vemurafenib on knockout #2 seems minimal. For knock out #1, vemurafenib does seem to further impair the growth.

The bar graph, which is shown below and provided as supplementary data in the revised version of our manuscript, highlights the significance of the additive effect of vemurafenib in resistant cells on a SREBP deficient background.

E
Figure S14 Proliferation of CRISPR-Cas9 knockouts of SREBF1 in combination with vemurafenib (5 μ M) treatment. Data was taken from the experiment shown in main figure 3E (time point 144h).

Fig S17-19: This is not particularly convincing. It appears that A101D cells are only partially resistant to dabrafenib/trametinb and there is very little effect of fatostatin. The D10 BMR cell line is much more resistant and shows a small effect of fatostatin but no real effect of betulin. Overall, it's hard to interpret these data.

A101D are indeed less resistant to dabrafenib/trametinb than D10BMR. For the sake of clarity we now do state this in the manuscript. We would like to point out that fatostatin and betulin inhibit SREBP through different mechanisms, which may explain their divergent effects in the D10BMR cell line.

However, as illustrated by the bar graphs that are shown below and presented as supplemental data in the revised version of the manuscript, both SREBP inhibitors significantly sensitise cells to BRAF targeting therapy.

Figure S19

(A) Proliferation curves of the dabrafenib + trametinib double resistant cell line A101D BMR treated with dabrafenib, trametinib, betulin and fatostatin (phase contrast density measured by the Incucyte system). (B) Bar graph of time point 120h.

Figure S21

(A) Proliferation curves of the dabrafenib + trametinib double resistant cell line D10 BMR treated with dabrafenib, trametinib, betulin and fatostatin (phase contrast density measured by the Incucyte system). (B) Bar graph of time point 192h for the fatostatin experiment, 108h for the betulin experiment.

Fig 4D: These expts claim to show that antioxidants partially rescue cell proliferation from combined SREBP-1/vemurafenib inhibition. This effect is quite minimal. Is there a way to express a p value?

To address this point, we have included bar graphs illustrating that AOX treatment significantly rescues cell proliferation following combined SREBP and BRAF inhibition (see also below).

Figure S27

Alpha-tocopherol (100 µM), ferrostatin (1.25 µM) or NAC (120 µM) were added to the culture medium of 451lu R cells and were treated with either vemurafenib (5 µM) alone or a combination with fatostatin or betulin. Data was taken from the experiment shown in main figure 4D (time point 192h).

We are grateful to this reviewer for his/her very useful comments.